# Increased mitochondrial protein import and cardiolipin remodelling upon *early* mtUPR

Daniel Poveda-Huertes[1], Asli Aras Taskin[1,2], Ines Dhaouadi[1], Lisa Myketin[1], Adinarayana Marada[1], Lukas Habernig[3], Sabrina Büttner[3,4], F.-Nora Vögtle[1,2¤]*

**1** Institute of Biochemistry and Molecular Biology, ZBMZ, Faculty of Medicine, University of Freiburg, Freiburg, Germany, **2** CIBSS—Centre for Integrative Biological Signalling Studies, University of Freiburg, Freiburg, Germany, **3** Department of Molecular Biosciences, The Wenner-Gren Institute, Stockholm University, Stockholm, Sweden, **4** Institute of Molecular Biosciences, University of Graz, Graz, Austria

☯ These authors contributed equally to this work.
¤ Current address: Center for Molecular Biology of Heidelberg University (ZMBH), Heidelberg, Germany
* n.voegtle@zmbh.uni-heidelberg.de

**Data Availability Statement:** All relevant data are within the manuscript and its Supporting Information files.

## Abstract

Mitochondrial defects can cause a variety of human diseases and protective mechanisms exist to maintain mitochondrial functionality. Imbalances in mitochondrial proteostasis trigger a transcriptional program, termed mitochondrial unfolded protein response (mtUPR). However, the temporal sequence of events in mtUPR is unclear and the consequences on mitochondrial protein import are controversial. Here, we have quantitatively analyzed all main import pathways into mitochondria after different time spans of mtUPR induction. Kinetic analyses reveal that protein import into all mitochondrial subcompartments strongly increases early upon mtUPR and that this is accompanied by rapid remodelling of the mitochondrial signature lipid cardiolipin. Genetic inactivation of cardiolipin synthesis precluded stimulation of protein import and compromised cellular fitness. At late stages of mtUPR upon sustained stress, mitochondrial protein import efficiency declined. Our work clarifies the enigma of protein import upon mtUPR and identifies sequential mtUPR stages, in which an early increase in protein biogenesis to restore mitochondrial proteostasis is followed by late stages characterized by a decrease in import capacity upon prolonged stress induction.

## Author summary

Mitochondria are essential organelles and involved in numerous important functions like ATP production, biosynthesis of metabolites and co-factors or regulation of programmed cell death. To fulfill this plethora of different tasks, mitochondria require an extensive proteome, which is build by import of nuclear-encoded precursor proteins from the cytosol. Mitochondrial defects can cause a variety of severe human disorders that often affect tissues with high energy demand e.g. heart, muscle or brain. However, protective mechanisms exist that are triggered upon mitochondrial dysfunction: Imbalances in mitochondrial proteostasis are sensed by the cell and elicit a nuclear transcriptional response, termed mitochondrial unfolded protein response (mtUPR). Transcription of mitochondrial chaperones and proteases is increased to counteract mitochondrial

**Funding:** This work was supported by grants from the Deutsche Forschungsgemeinschaft (DFG) under Germany's Excellence Strategy (CIBSS - EXC-2189 - Project ID 390939984; to F.N.V.), the SFB 1381 (Project-ID 403222702; to F.N.V.), the Emmy-Noether Programm (to F.N.V.) and the RTG 2206/423813989 (to F.N.V.), the Swedish Research Council Vetenskapsrådet (VR; 2019-05249; to S.B.), the Knut and Alice Wallenberg foundation (2017.0091; to S.B.), the Austrian Science Fund FWF/P27183-B24 (to S.B.) and the Stiftelsen Olle Engkvist Byggmästare (OEB; 194-0681 and 207-0527; to S.B.) The funders had no role in study design, data collection and analysis, decision to publish, or preparation of the manuscript.

**Competing interests:** The authors declare that they have no conflict of interest.

dysfunctions. In this study, we investigated if mtUPR progresses in different temporal stages and how protein import is affected upon mtUPR. We discover that mtUPR is subdivided into an early phase, in which protein import increases and a late phase, in which it declines. Stimulation of protein import is accompanied by an increase and remodelling of the mitochondrial signature lipid cardiolipin. Our work establishes a novel model how cells respond to dysfunctional mitochondria, in which cardiolipin and protein import are modulated as first protective measures.

## Introduction

Mitochondria are essential organelles and dysfunctions can cause severe human disorders [1–4]. Imbalances in mitochondrial proteostasis elicit a protective transcriptional program, termed mitochondrial unfolded protein response (mtUPR) that has been described in various model organisms from yeast to mouse and human tissue culture [5–9]. A common characteristic of mtUPR across species is the increased transcription of genes encoding mitochondrial proteases and chaperones to counteract the defects in mitochondrial proteostasis [6,7,9–12]. Current models of mtUPR are mainly based on the assumption that mitochondrial defects result in impaired protein import. In *C. elegans*, this block in mitochondrial protein import is reported to trigger re-localization of the transcription factor ATFS-1 from mitochondria to the nucleus [8,13,14]. The subsequent reprogramming of nuclear gene expression to restore mitochondrial proteostasis is well-established, but it has remained enigmatic how the newly synthesized mitochondrial chaperones and proteases can be imported into import-incompetent mitochondria [5,8]. We recently established the transcription factor Rox1 as crucial for cellular survival upon mtUPR in yeast [15]. Intriguingly, just as ATFS-1 Rox1 changes its subcelluar distribution upon mtUPR, but in the opposite direction: Usually localized to the nucleus, Rox1 re-translocates to and is imported into mitochondria upon mtUPR, where it binds to mitochondrial DNA (mtDNA), thereby maintaining mitochondrial transcription and translation. This protective function of Rox1 depends on its efficient import into mitochondria raising the exciting hypothesis that mtUPR proceeds in different temporal stages: early phases in which protein import competence into mitochondria is maintained and late stages that are characterized by a decrease in protein import. However, this hypothesis challenges the current paradigm that impaired protein import is an essential prerequisite for mtUPR induction.

Although mitochondria contain their own genome, they strictly depend on import of precursor proteins from the cytosol as 99% of mitochondrial proteins are encoded in the nuclear DNA [16–19]. So far, analysis of protein import upon mtUPR focused solely on the presequence import pathway (TOM-TIM23/PAM) into the matrix [14,20,21]. However, mitochondrial protein biogenesis is a complex process in which several sophisticated import machineries cooperate and functional mitochondria can only be maintained if all—membrane potential-dependent and -independent—import pathways into the four different mitochondrial subcompartments function properly (Fig 1A; [16–19]). Furthermore, mostly protein steady state levels were analyzed to assess import competence [14,20,21], which does not allow comparison of the actual import rates. The role of protein biogenesis upon mtUPR therefore remains elusive, as a comprehensive kinetic analysis of the different import pathways into mitochondria is lacking.

Here, we have for the first time analyzed all major import routes into mitochondria at different times after mtUPR induction. The use of authentic (non-tagged) substrates and kinetic analysis of their import and assembly pathways reveal the surprising finding that protein

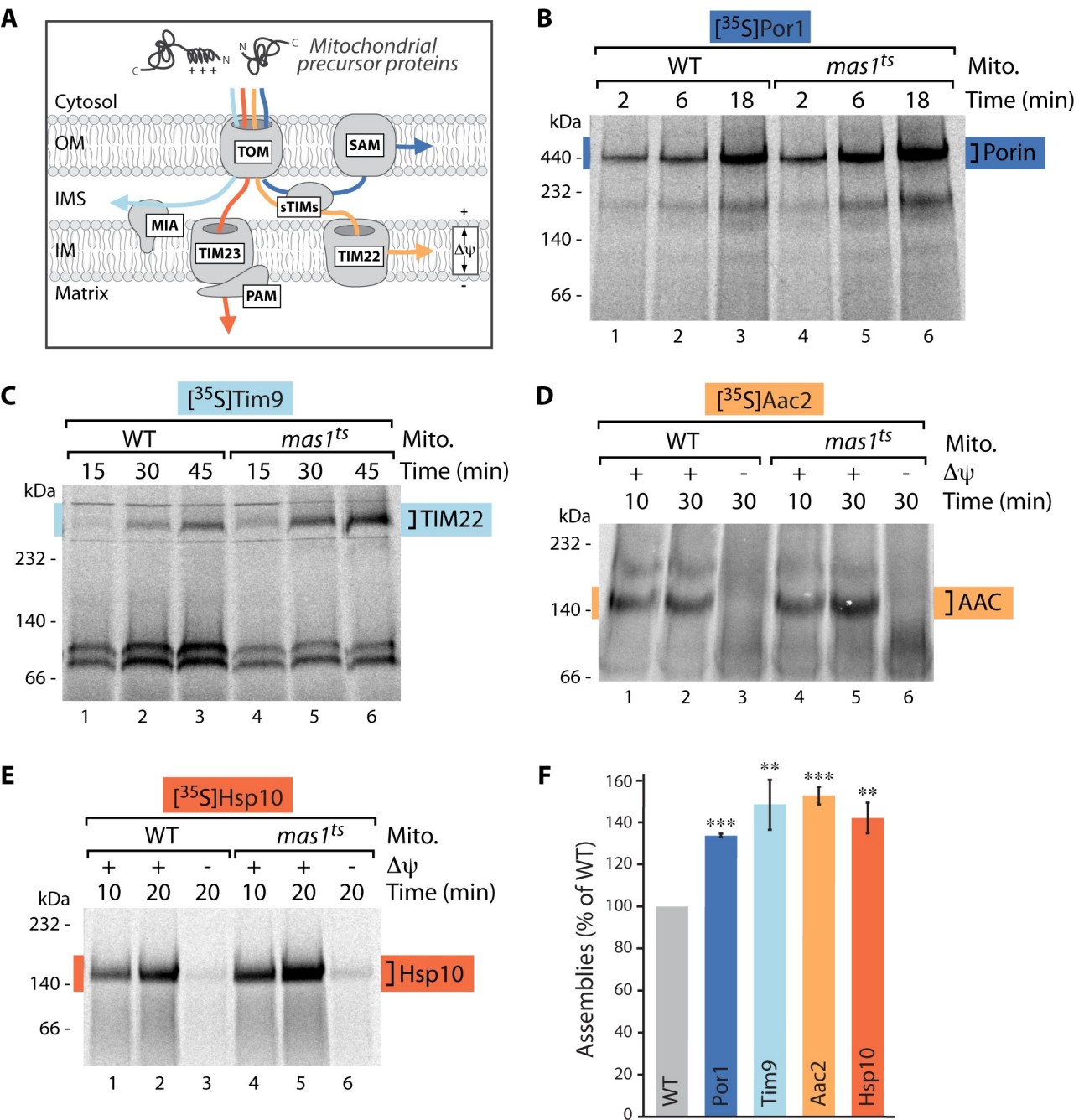

**Fig 1. Protein import into all mitochondrial subcompartments is stimulated upon mtUPR.** (A) Schematic of the main protein import pathways into mitochondria. Protein translocation into the matrix and inner membrane depends on the membrane potential (Δψ) as driving force, protein import into the outer membrane or intermembrane space is membrane potential-independent. TOM, translocase of the outer membrane; SAM, sorting and assembly machinery; sTIMs, small TIMs; MIA, mitochondrial intermembrane space assembly; TIM23 and TIM22, translocases of the inner membrane; PAM, presequence translocase associated motor; OM, outer mitochondrial membrane; IMS, intermembrane space; IM, inner mitochondrial membrane. (B)-(E) Import kinetics of indicated radiolabeled precursor proteins into mitochondria isolated from wild-type (WT) or *mas1^{ts}* cells after induction of mtUPR for 10 hours. Where indicated (in case of matrix- and inner membrane-targeted substrates), the membrane potential (Δψ) was dissipated prior to the import reaction. Samples were solubilized in the mild detergent digitonin and analyzed by BN-PAGE and autoradiography. In Fig 1C the soluble Tim9/Tim10 complex is formed at appr. 100 kDa. (F) Quantification of longest import time-point displayed in (B)-(E) normalized to WT value. n = 3, data represent means ± SEM. Student's t-test was used for pairwise comparison. **p < 0.01; ***p < 0.001.

import is not inhibited, but on the contrary strongly enhanced during early stages of mtUPR. Only upon prolonged stress, import rates into mitochondria decline. Systematic lipid profiling identified increased synthesis and rapid remodelling of the mitochondrial signature lipid cardiolipin (CL) early upon mtUPR induction. The resulting changes in CL species surrounding the import machineries likely stabilize the translocases and stimulate protein import. Consequently, genetic inactivation of CL synthesis results in decreased import rates already during early mtUPR stages.

## Results

### Mitochondrial protein import is stimulated upon early mtUPR

To comprehensively investigate mitochondrial protein biogenesis upon mtUPR we analyzed import kinetics of all four major import routes into mitochondria by using genuine precursor proteins. To induce mtUPR and to identify possible different temporal stages we employed yeast cells harboring a temperature-sensitive variant of the mitochondrial processing protease MPP (point mutation in the catalytic subunit Mas1, *mas1*$^{ts}$ [15,22]). MPP is the major presequence protease and cleaves off the N-terminal targeting signals from incoming precursor proteins in the mitochondrial matrix [16,17,19,23,24]. The conditional MPP mutant *mas1*$^{ts}$ grows indistinguishable from wild-type cells at permissive temperature (S1A Fig). Upon growth at elevated temperature MPP is inactivated (S1B Fig) and unprocessed precursor proteins accumulate in the matrix [15]. The precursors aggregate and trigger a rapid transcriptional response already two hours after MPP impairment that induces expression of mitochondrial chaperones and proteases [15]. Importantly, the mitochondrial membrane potential is not affected in this mtUPR model [15].

To assess the import competence of mitochondria upon different temporal stages of mtUPR, mitochondria from wild-type and *mas1*$^{ts}$ cells were isolated after growth at permissive temperature (23˚C) or at different time points after induction of mtUPR (4, 10, 20 hours). The isolated organelles were then incubated with radiolabeled model substrate proteins of the TOM-SAM-, TOM-MIA-, TOM-TIM23/PAM- and TOM-TIM22-pathways, which represent the major protein import routes into mitochondria [16–19] (Fig 1A). While import into the outer membrane and intermembrane space does not depend on the membrane potential, protein translocation into the inner membrane and matrix requires the proton gradient across the inner membrane as driving force. We monitored import kinetics to directly compare the mitochondrial import competence and used blue-native PAGE analysis, which additionally enables assessment of the capacity of the imported substrates to assemble into mature functional complexes [25]. Import into mitochondria isolated after growth at permissive conditions did not reveal differences between wild-type and the MPP mutant (S1C–S1H Fig). However, already after induction of mtUPR for only 4 hours, import into the inner membrane and matrix was stimulated as the precursors of the inner membrane carrier Aac2 and the soluble matrix protein Hsp10 showed enhanced assemblies in the *mas1*$^{ts}$ mutant (S2 Fig). This rather unexpected increase of mitochondrial import capacity was even more pronounced after induction of mtUPR for 10 hours: All four major import pathways, membrane potential-dependent and -independent, represented by the outer membrane proteins Por1, Tom40 and Tom22, the IMS protein Tim9, the inner membrane protein Aac2 and the matrix protein Hsp10, imported and assembled significantly better upon mtUPR (Figs 1B–1F and S3). Therefore, mitochondrial protein import is clearly increasing early upon mtUPR induction. These results are in opposition to previous studies proposing a general impairment of import upon mtUPR and current mtUPR models are mainly based on the assumption that protein import is blocked upon stress. However, most studies used strong, often irreversible stress triggers that lead to a loss of

membrane potential and consequently result in impaired membrane potential-dependent protein import into the matrix [14,26–29]. We hypothesized that physiologically relevant stress triggers might be milder, building up gradually over a longer period of time, and thus could result in a sequence of distinct mtUPR stages. Severe stress induction might preclude the initial, first phases of mtUPR, only occurring upon mild and incremental stress induction. As a consequence the use of severe stress triggers so far only allowed the detection and analyses of late mtUPR stages. In contrast, our conditional *mas1^ts^* mutant allows mild and reversible stress induction and therefore represents the ideal model to assess different temporal stages of mtUPR [15]. To test this hypothesis we next analyzed mitochondria isolated after induction of mtUPR for 20 hours (*late* mtUPR) and assessed protein import. Indeed, this prolonged stress clearly compromised import into outer membrane, inner membrane and matrix (S4A–S4D Fig). Importantly, this decrease in membrane potential-dependent import pathways (inner membrane and matrix) was not caused by a lower membrane potential after sustained mtUPR induction (S4E Fig).

A common characteristic of all tested substrate proteins is their strict dependence on mitochondrial import machineries. Only very few outer membrane proteins do not require the TOM and SAM translocases, e.g. the peripheral outer membrane receptor Tom20 [30]. Intriguingly, assembly of Tom20 was not changed in WT and *mas1^ts^* upon mtUPR induction for 10 hours (S5 Fig). Notably, the enhanced protein import upon mtUPR was not caused by upregulated transcription or increased protein steady state levels of import machinery subunits or by enhanced formation of the functional import complexes (S6 Fig).

Taken together, the conditional *mas1^ts^* mutant enables analysis of different temporal stages in mtUPR and mitochondrial protein import capacity upon mtUPR progression. Our data reveal that mtUPR indeed progresses in distinct chronological phases: During early mtUPR mitochondrial protein import is increased and only decreases in later stages. Thus, stimulation of protein import by modulation of the protein import machineries represents an early cellular response to mitochondrial dysfunction caused by defective MPP processing.

## mtUPR induces synthesis and remodelling of the mitochondrial signature lipid cardiolipin

As protein steady state levels of protein translocase subunits and assembly of the mature translocases were not changed upon mtUPR, we wondered about the underlying mechanism that stimulates protein import. Protein translocases are embedded in the lipid bilayer and phospholipids are the main building block of both mitochondrial membranes. The lipid composition influences the organellar shape, structure and function as membrane protein complexes including protein translocases critically depend on their lipid environment [31–35]. Changes in phospholipids could therefore be a mean to modulate the mitochondrial protein import machineries upon mtUPR. We performed lipidomic analysis of highly purified mitochondria [36] after short induction of mtUPR (0, 2 and 4 hours) to identify dynamic changes in the phospholipid composition of WT and *mas1^ts^* mitochondria (Fig 2A and 2B). While most phospholipids appeared unchanged in response to mtUPR, we observed dynamic changes in the mitochondrial signature lipid cardiolipin (CL): Whereas the total CL content was slightly decreased in WT, *mas1^ts^* mitochondria showed an increase in CL after 4 hours of mtUPR (Fig 2B). CL is a unique phospholipid with a dimeric structure formed by two phosphatidyl moieties that are linked by a glycerol bridge and contributes to the integrity of several mitochondrial protein complexes [31,37–40]. CL biosynthesis occurs in the inner membrane and starts with the synthesis of CDP-diacylglycerol (CDP-DAG) from phosphatidic acid. Strikingly, when analyzing the CDP-DAG precursor, *mas1^ts^* cells showed a strong increase after 2 hours,

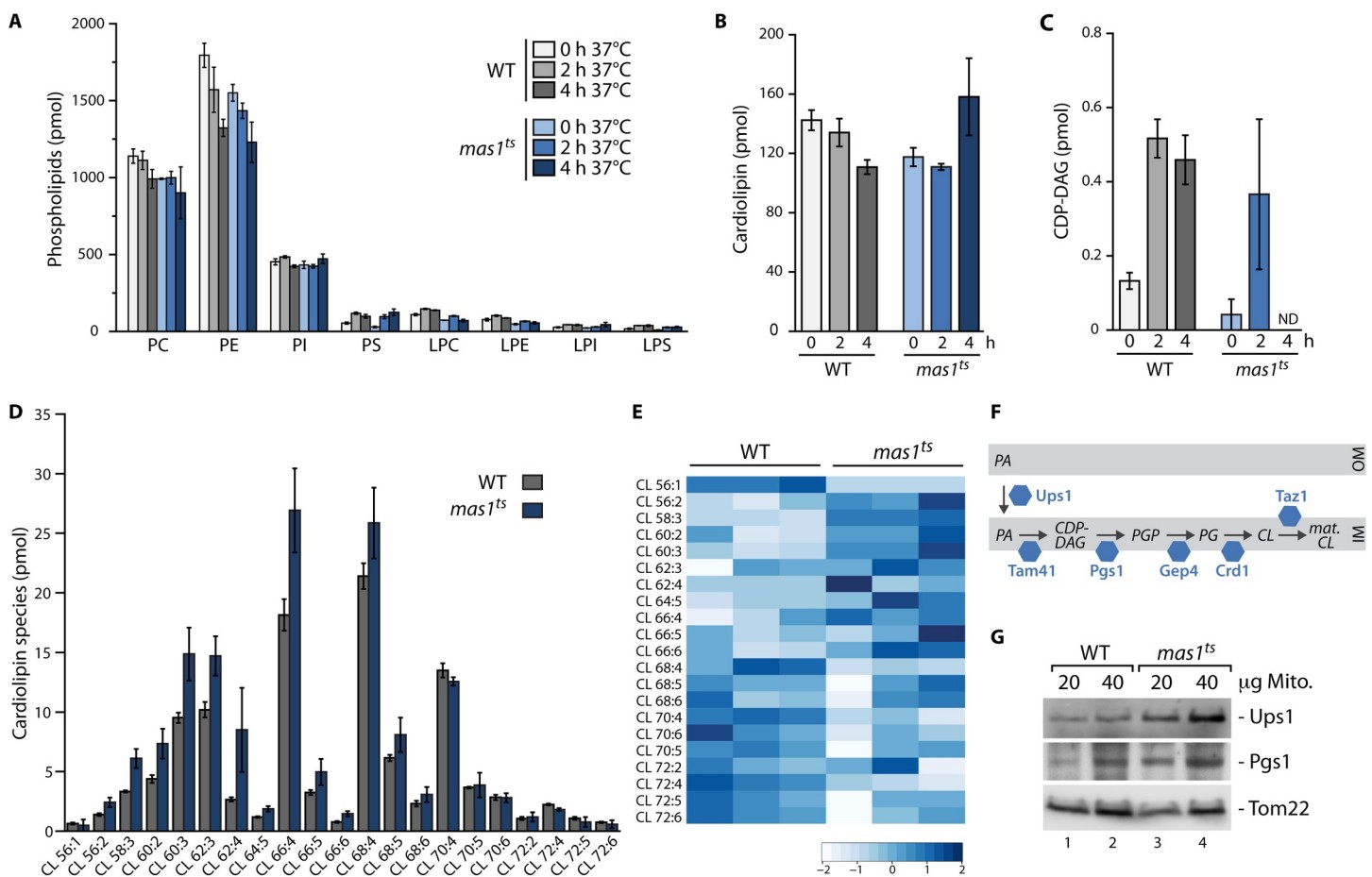

**Fig 2. mtUPR induces dynamic remodelling of the mitochondrial signature lipid cardiolipin.** (A) Lipidomic analysis of indicated phospholipids in highly purified wild-type (WT) and *mas1ᵗˢ* mitochondria isolated after growth at permissive temperature, or after induction of mtUPR for 2 and 4 hours. n = 3, data represent means ± SEM. PC, phosphatidylcholine; PE, phosphatidylethalonamine; PI, phosphatidylinositol; PS, phosphatidylserine; LPC, lysophosphatidylcholine; LPE, lysophosphatidylet.ethalonamine; LPI, lysophosphatidylinositol; LPS, lysophosphatidylserine. (B) Quantification of cardiolipin in WT and *mas1ᵗˢ* mitochondria isolated after growth under non-stress conditions, or after mtUPR induction for 2 or 4 hours. Non-standardized, absolute values in pmol measured in total lipid extracts from isolated, highly purified mitochondria are shown. n = 3, data represent means ± SEM. (C) Quantification of the CL-precursor CDP-diacylglycerol (CDP-DAG) in WT and *mas1ᵗˢ* mitochondria isolated after growth at permissive temperature, or after induction of mtUPR for 2 and 4 hours. Non-standardized, absolute values in pmol measured in total lipid extracts from isolated, highly purified mitochondria are shown. n = 3, data represent means ± SEM. ND, not detected. (D) Quantification of indicated CL subspecies in WT and *mas1ᵗˢ* mitochondria isolated after induction of mtUPR for 4 hours. Non-standardized, absolute values in pmol measured in total lipid extracts from isolated, highly purified mitochondria are shown. n = 3, data represent means ± SEM. (E) Heatmap of distribution of different CL subspecies standardized to total CL content per sample in WT and *mas1ᵗˢ* mitochondria isolated from cells shifted for 4 hours to non-permissive growth temperature. Heatmaps were generated using standardized values in mol% and thus total CL content in each individual sample was set to 100% to represent relative CL species distribution within each sample. Relative changes, scaled and centered for each CL species, are depicted. Shown are three biological replicates for each strain. (F) Schematic overview of the CL biosynthesis pathway in mitochondria. Adapted from [41]. PGP, phosphatidylglycerolphosphate; PG, phosphatidylglycerol; OM, outer mitochondrial membrane; IM, inner mitochondrial membrane. (G) Immunoblot analysis of WT and *mas1ᵗˢ* mitochondria isolated from cells shifted to non-permissive temperature for 10 hours. Tom22 serves as loading control.

while it was not detectable at all after 4 hours (Fig 2C). This suggests that cells increase the biosynthesis of CL upon mtUPR and rapidly convert its precursor into CL. The four acyl chains of CL vary in length and saturation and analysis of CL subspecies revealed that mitochondria remodel the CL species upon mtUPR induction, resulting in a shift towards shorter acyl chains (Figs 2D, 2E and S7A). This remodelling dynamically continues so that after 10 hours of mtUPR induction CL subspecies between wild-type and the *mas1ᵗˢ* differ in acyl chain length and saturation (S7B–S7D Fig).

To investigate the mechanism behind the changes in CL levels, we analyzed the steady state levels of proteins involved in cardiolipin biosynthesis and detected an increase in Ups1 and Pgs1 in the *mas1*$^{ts}$ mutant after mtUPR induction (Fig 2F and 2G). Ups1 plays a crucial role in CL metabolism by transporting the precursor phosphatidic acid from the outer to the inner mitochondrial membrane, where CL lipid synthesis takes place [38]. Pgs1 (phosphatidylglycerolphosphate synthase) catalyzes the synthesis of phosphatidylglycerolphosphate (PGP) from CDP-DAG, which is the first and rate-limiting step of CL biosynthesis [38]. Notably, CDP-DAG was not detectable in *mas1*$^{ts}$ mitochondria after 4 hours of mtUPR induction (Fig 2C), which could point towards rapid turnover of CDP-DAG by increased Pgs1.

In summary, cells respond to mtUPR induction with increased synthesis and dynamic remodelling of the mitochondrial signature lipid cardiolipin, which is likely induced by elevated abundance of Ups1 and Pgs1, two proteins critically involved in CL biosynthesis.

## mtUPR-induced CL modulation changes the lipid environment surrounding the protein translocases

To further study a possible role of lipids in mtUPR-induced import stimulation on a genetic level, we deleted several genes involved in phospholipid biosynthesis in WT and *mas1*$^{ts}$ cells and assessed growth at permissive temperature (23°C) and upon mild induction of mtUPR (35°C). Notably, deletion of the genes coding for the cardiolipin synthase, Crd1 and the lyso-phosphatidylcholine acyltransferase, Taz1, required for CL remodelling, resulted in a severe growth defect upon induction of mtUPR on solid and liquid growth medium (Fig 3A–3D). In contrast, deletion of genes encoding enzymes within the biosynthesis pathways of all other phospholipids (phosphatidic acid, phosphatidylserine, phosphatidylethanolamine or phosphatidylcholine) had no effect on growth of the *mas1*$^{ts}$ cells (S8A Fig).

While these data indicate that both cardiolipin synthesis and remodeling is required to maintain cellular fitness upon mtUPR, we next wondered if the changes in CL species might also affect the import translocases and their activity upon mtUPR. CL is most abundant in the inner membrane and has been proposed to promote clustering of membrane proteins and to stabilize the respiratory chain complexes [31,41]. To assess changes in the lipid environment surrounding the inner membrane translocases, we tested their extraction from the lipid bilayer by solubilization of isolated mitochondria with increasing concentrations of the mild detergent digitonin. Both translocases, the TIM22 and the TIM23 complex, showed a different extraction profile when comparing wild-type and *mas1*$^{ts}$ mitochondria that had been exposed to elevated temperature for 4 or 10 hours, respectively (Figs 3E and S8B). Efficient extraction of both translocases from their endogenous lipid environment required higher amounts of digitonin in *mas1*$^{ts}$. The protein steady state levels of mitochondrial import machinery subunits were not changed under these conditions (S6B and S8C Figs). This indicates that the lipid composition surrounding the import machineries changes upon mtUPR. When we performed the same analysis in mitochondria isolated from cells that in addition lacked the cardiolipin synthase Crd1, no differences in the extraction profiles between *mas1*$^{ts}$ and control were detectable (S8D Fig).

Taken together, cardiolipin synthesis and remodelling is essential for cellular fitness upon mtUPR and this mtUPR induced CL modulation changes the embedding of the translocases in the lipid bilayer.

## Cardiolipin is required to maintain protein import upon mtUPR

We next assessed if modulation of cardiolipin upon mtUPR and the concomitant changes in the lipid environment around the translocases are a requirement for the observed stimulation of protein import upon mtUPR. Hence, we analyzed import kinetics of radiolabeled model

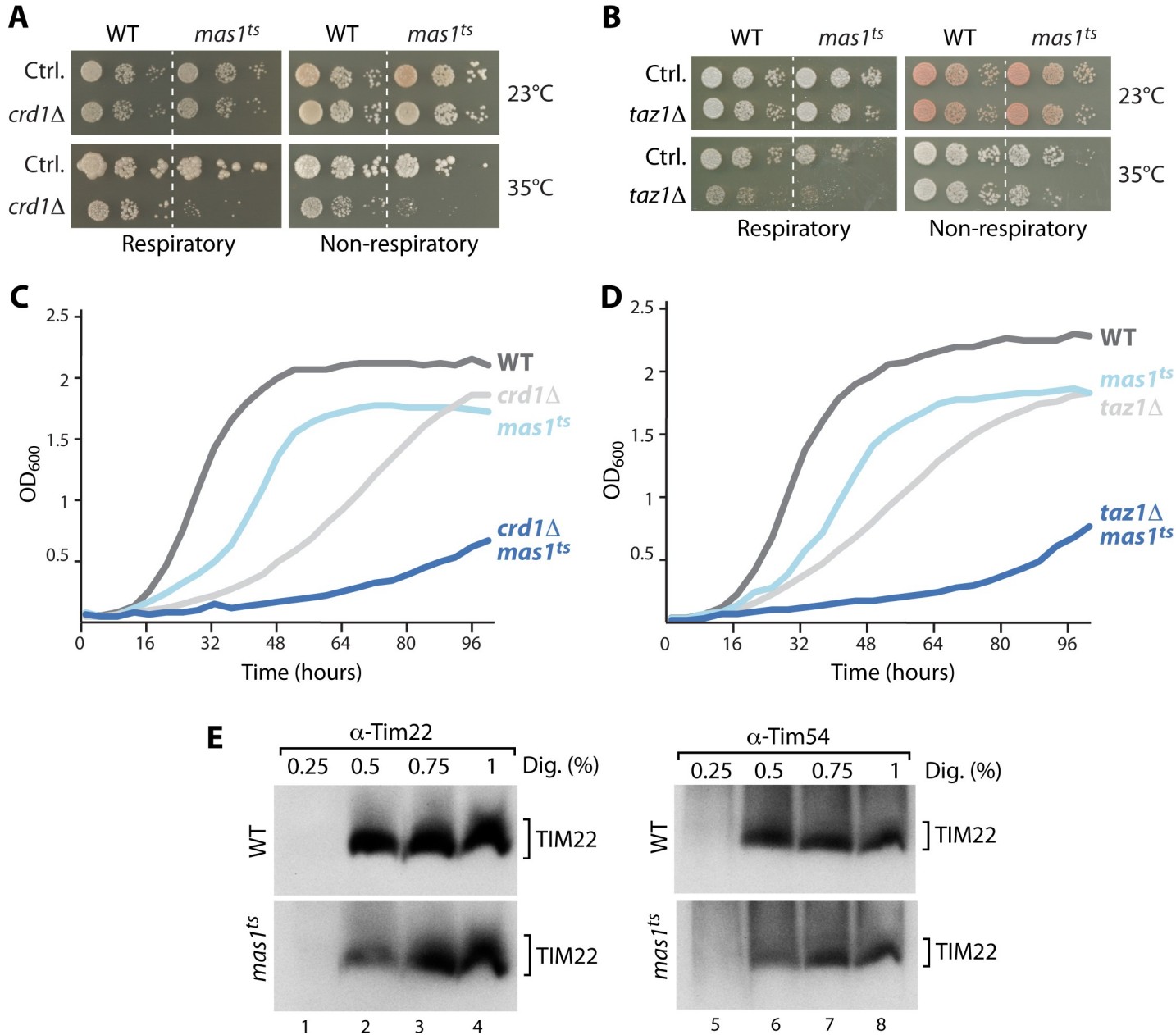

**Fig 3. Cellular growth and translocase extractability depend on CL modulation.** (A) and (B) Growth assay to test for synthetic effects of *CRD1* or *TAZ1* deletion in wild-type (WT) and *mas1^ts* cells. Serial dilutions were tested on respiratory (YPglycerol) and non-respiratory (YPglucose) plates and incubated at 23°C or 35°C, respectively. (C) and (D) Growth curves of indicated yeast cells on respiratory growth medium (YPglycerol) at 35°C. (E) Analysis of native protein complexes in WT and *mas1^ts* mitochondria isolated from cells grown for 4 hours at non-permissive temperature. Samples were solubilized with indicated concentrations of digitonin and analyzed by Blue-native PAGE and immunodecoration. TIM22, translocase of the inner mitochondrial membrane.

precursor proteins into mitochondria isolated from *crd1Δ* and *crd1Δmas1^ts* cells that had been grown for 10 hours at elevated temperature to induce mtUPR. While the efficiency of all import pathways was strongly increased in the *mas1^ts* mutant compared to WT at this early time of mtUPR induction (Figs 1B–1F and S2), the absence of the cardiolipin synthase resulted in a severe loss of this stimulation. Protein import into the outer membrane, inner membrane and the matrix was significantly decreased in the absence of Crd1 (Fig 4A–4D). Notably,

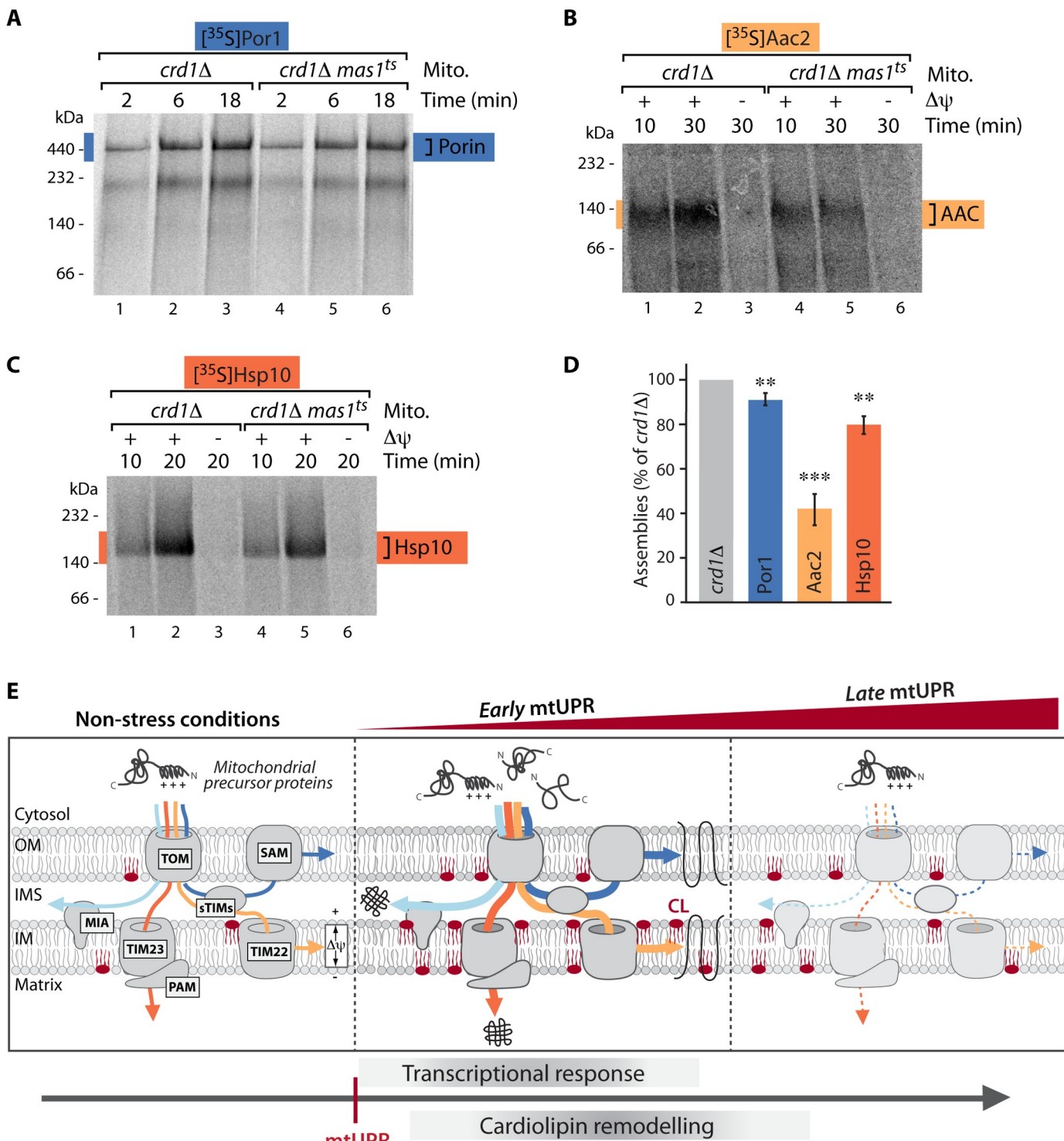

**Fig 4. Cardiolipin is required to maintain protein import upon mtUPR.** (A)—(C) Radiolabeled Por1 (A), Aac2 (B) and Hsp10 (C) precursors were imported for the indicated times into *crd1Δ* and *crd1Δmas1ts* mitochondria isolated after induction of mtUPR for 10 hours. Where indicated the membrane potential (Δψ) was dissipated prior to the import reaction. Assembled complexes were analyzed via autoradiography after Blue Native PAGE. (D) Quantification of imports from (A)-(C). Compared is the longest import time point from *crd1Δmas1ts* in comparison to *crd1Δ*. n = 3, data represent means ± SEM. Student's t-test was used for comparison. **p < 0.01; ***p < 0.001. (E) Model of early and late events upon mtUPR. Increased cardiolipin (CL) upon early mtUPR results in stabilization of translocases in the outer and inner mitochondrial membrane and boosts mitochondrial protein import into all compartments (*early* mtUPR). Protein import decreases only upon prolonged stress (*late* mtUPR).

neither the protein steady state levels of mitochondrial import components nor the membrane potential across the inner membrane was affected (S9A and S9B Fig).

We conclude that an increase in CL and a rearrangement of CL subspecies is triggered upon mtUPR. Furthermore, lack of CL and by this lack of CL re-arrangement precludes stimulation of mitochondrial protein import upon mtUPR.

## Discussion

Our findings demonstrate that mtUPR progresses in consecutive temporal stages. In the first phase of the response (*early* mtUPR) a rapid remodelling of the mitochondrial signature lipid cardiolipin is accompanied by a boost of protein import into all mitochondrial subcompartments. This coincides with the well-established reprogramming of nuclear gene expression upon mtUPR that induces the transcription of mitochondrial chaperone and protease genes to restore mitochondrial proteostasis [15]. Modulation of CL likely stabilizes the mitochondrial translocases to ensure that these newly synthesized protective proteins indeed reach their destination within the organelle. Only in later phases upon persistence of mitochondrial dysfunction caused by defective MPP processing (*late* mtUPR), protein import into mitochondria decreases.

Our study represents the first systematic analysis of mitochondrial protein biogenesis upon mtUPR, using authentic substrates and import kinetics to assess the capacity of membrane potential-dependent and -independent import pathways. While these extensive analyses revealed that all protein import pathways are enhanced upon mtUPR, previous publications proposed a general block of import upon mtUPR, which is also the basis of current mtUPR models [14,42]. However, previous analyses focused on the presequence import pathway into the matrix and were mainly relying on non-authentic GFP fusion proteins and analysis of protein steady state levels instead of kinetics, precluding direct assessment of import rates [14,21]. A further difference between our present analyses and previous studies are the triggers applied to elicit mtUPR. Former studies used strong and often irreversible stress induction, e.g. inhibitors of the respiratory chain or depletion of mtDNA [5]. Several of these treatments effectively dissipate the membrane potential across the inner membrane, which is the driving force for protein import into the matrix and inner membrane. Consequently, protein import via the presequence pathway is abrogated under these stress conditions. While screening of various stress triggers implicated a reduced membrane potential as general signal for mtUPR induction [26], the question if impaired import is a consequence or prerequisite for mtUPR induction remained unanswered. In contrast to previously used, severe stress triggers, the mild induction of mtUPR in the conditional *mas1^{ts}* mutant does not affect the membrane potential and thereby allows capturing and investigation of the early and late phases in mitochondrial stress responses. The use of the temperature-sensitive cells clearly reveals that import is actively stimulated in early mtUPR phases induced by growth at non-permissive temperature and only declines in late stages upon prolonged stress. Import under permissive growth conditions was not affected, showing that the increased import capacity in *mas1^{ts}* represents a specific response to mtUPR induction.

We therefore propose that cells respond to mitochondrial dysfunctions in successive steps: As first line of defense a nuclear transcriptional response is triggered that coincides with CL remodelling to boost import of newly synthesized mitochondrial proteins. Also the nuclear transcription factor Rox1, which is required to maintain mitochondrial transcription and translation and is essential for cellular survival upon mtUPR, is imported into mitochondria at this early stage [15]. Upon prolonged stress or upon severe mitochondrial damage, import into mitochondria declines, which then triggers translocation of transcription factors like ATFS-1 into the nucleus as proposed in current mtUPR models [8,14,11].

The stimulated protein import upon early mtUPR is accompanied by the modulation of cardiolipin. The increased abundance and different acyl chain composition of cardiolipin species seems to result in a changed lipid microenvironment around the translocases that stabilizes the protein import machineries upon mtUPR. Furthermore, it was shown that precursors bind efficiently to membranes with a high CL content [43,44]. This was suggested to support sequestration of proteins in a translocation-competent conformation by inducing a partial unfolding of the precursor [45,46] and could therefore further stimulate protein import under stress conditions. Further studies will reveal the impact of these different aspects of CL modulation on import efficiency and shed light on the regulation of the different temporal mtUPR stages. Furthermore, an in-depth analysis of the sequence of events in mitochondrial stress responses may define time windows to therapeutically modulate the response.

## Material and methods

### Yeast cells and growth conditions

YPH499 (Mata, ade2-101, his3-Δ200, leu2-Δ1, ura3-52, trp1-Δ63, lys2-801) is the parental *Saccharomyces cerevisiae* strain used to generate all mutants of this study [47] (see S1 Table for all strains used in this study). The strain *mas1^ts* (R144C) is a temperature-sensitive mutant that was described previously in [22]. The nourseothricin cassette (natNT2) was employed to create deletion strains by homologous recombination of the gene of interest [48,49].

Yeast cells were grown in YPG medium (1% [w/v] yeast extract, 2% [w/v] bacto peptone, 3% [w/v] glycerol, pH 4.9) or YPD medium (1% [w/v] yeast extract, 2% [w/v] bacto peptone, 2% [w/v] glucose, pH 4.9) at 23°C (permissive conditions), 35°C or 37°C (non-permissive conditions, mtUPR). For early mtUPR analysis cells were shifted for 2, 4 or 10 hours to 37°C prior to isolation of mitochondria. For late mtUPR analysis cells were grown for 20 hours at non-permissive temperature. The optical density (OD) of the cell culture was measured at a wavelength of 600 nm ($OD_{600}$). For growth tests on solid medium, tenfold serial dilutions were spotted on YPD or YPG agar plates. The plates were incubated at different temperatures. Liquid growth curves were obtained by measuring the absorbance at 600 nm with a 48 well plate reader (BMG LABTECH Reader CLARIOstar) (measurement every 5 min). An initial $OD_{600}$ of 0.2 was inoculated in YPG medium. YPG medium without cells was used as blank.

### Isolation of mitochondria

Isolation of mitochondria from *S. cerevisiae* was performed according to established protocols [36]. Yeast cells grown on respiratory medium (YPG) at permissive or non-permissive temperatures were harvested by centrifugation. The yeast pellet was treated with dithiothreitol (DTT) buffer (10 mM DTT, 100 mM Tris-$H_2SO_4$, pH 9.4) and subsequently with zymolyase buffer (1.2 M sorbitol, 20 mM $K_2HPO_4$-HCl, pH 7.4) containing 3 mg zymolyase per gram of yeast cells. Obtained spheroplasts were washed and then disrupted by homogenization. After the homogenization, mitochondria were isolated by differential centrifugation. The mitochondrial pellet was resuspended in SEM buffer (250 mM sucrose, 1 mM EDTA, 10 mM MOPS-KOH, pH 7.2). The concentration of the mitochondria was determined by Bradford assay and adjusted to a concentration of 10 mg/mL with SEM buffer. Isolated mitochondria were aliquoted, snap-frozen in liquid nitrogen and stored at -80°C.

### Sucrose gradient purification of isolated mitochondria

Highly purified mitochondria for lipidomic analyses were obtained as previously described [36]. Crude mitochondrial fractions were adjusted to a concentration of 5 mg/mL with SEM

buffer and homogenized with a glass potter by 10 strokes. The homogenate was loaded on top of a sucrose gradient and centrifuged for 1 h at 2°C and 100.000 x g. The purified mitochondria were recovered with a Pasteur pipet from the 60–32% sucrose interface. Collected purified mitochondrial fraction was diluted in SEM and centrifuged at 10.000 x g at 2°C for 15 min. The mitochondrial pellet was suspended in SEM buffer and the protein concentration was determined by Bradford assay. Purified mitochondrial samples were snap-frozen in liquid nitrogen and stored at -80°C.

## Analysis of mitochondrial proteins by SDS-PAGE, BN-PAGE and immunoblotting

Isolated mitochondria were washed in SEM buffer (250 mM sucrose, 1 mM EDTA, 10 mM MOPS-KOH, pH 7.2) and centrifuged at 20 000 x g for 10 min at 4°C. Mitochondrial pellets were resuspended in Laemmli buffer containing 2% [w/v] SDS, 10% [v/v] glycerol, 0.02% [w/v] bromophenol blue, 62.5 mM Tris-HCl, pH 6.8, 1% [v/v] β-mercaptoethanol and incubated for 15 min at 65°C. Samples were analyzed by SDS-PAGE and immunodecoration according to standard protocols.

Native protein complexes were analyzed by BN-PAGE according to established protocols [25]. 50 μg mitochondria were solubilized in digitonin solubilization buffer (1% [w/v] digitonin, 20 mM Tris-HCl, pH 7.4, 0.5 mM EDTA, 10% [v/v] glycerol, 50 mM NaCl) and incubated on ice for 15 min. To assess extraction of translocases from the lipid bilayer, mitochondria were solubilized in solubilization buffer containing varying digitonin concentrations (0.25, 0.5, 0.75 and 1% [w/v]). Solubilized mitochondrial samples were centrifuged at 20 000 x g for 5 min at 4°C. The supernatant was analyzed on 4–13% or 4–16.5% gradient BN-PAGE followed by immunoblotting. All antisera used in this study are listed in S2 Table.

## In organello import of radiolabeled precursor proteins

Radiolabeled precursor proteins were synthesized in vitro using the rabbit reticulocyte lysate system (Promega) in the presence of radioactive $^{35}$S-methionine. For the precursors Aac2 and Tim9 the transcription/translation system TNT Coupled Reaction Mix (Promega) was used. Isolated mitochondria (20–70 μg) were resuspended in import buffer (3% [w/v] bovine serum albumin (BSA), 250 mM sucrose, 80 mM KCl, 5 mM L-methionine, 5 mM $MgCl_2$, 2 mM $KH_2PO_4$, 10 mM MOPS-KOH, pH 7.2, supplemented with 2 mM NADH and 2 mM ATP). Import buffer without BSA and NADH was used for Tim9 assembly. Resuspended mitochondria were pre-incubated for 2 min at the desired temperature and then import and assembly reactions were started by addition of radiolabeled precursors. The samples were incubated for different time points at 20°C (Por1) or at 25°C (Tim9, Aac2, Hsp10, Tom40, Tom22, Tom20). Where indicated, the mitochondrial membrane potential was disrupted preceding the import reaction by addition of AVO (8 μM antimycin A, 1 μM valinomycin and 20 μM oligomycin). The import reactions were stopped by placing the samples on ice. For import pathways that depend on the membrane potential, addition of AVO was used to terminate the import reaction. Mitochondria were washed with SEM buffer. To analyze protein import and assembly, samples were lysed in solubilization buffer with the appropriate amount of digitonin. For Tom22, Tom40, Tim9, Aac2 and Hsp10 assemblies a concentration of 1% [w/v] digitonin was used for solubilization. Por1 and Tom20 required a concentration of 0.3% [w/v] digitonin. Protein complexes were separated on a blue native gradient gel (BN-PAGE) (4–12% for TOM, TIM22 and Porin and 6–16% for AAC and Hsp10 complexes) and further analyzed by digital autoradiography.

## RTqPCR analysis

Total RNA was isolated using the RNeasy Mini kit (Qiagen) and reverse transcribed (Applied biosystems reagents). Quantitative PCR reactions were performed using the SsoAdvanced Universal SYBR Green Supermix (Bio-Rad) and the following primers: FwTaf10, 5' TCCT CCTATCATTCCCGATGC 3'; RvTaf10, 5' CGCTACGGAAGACCTGATCC 3'; FwIrc25, 5' ACGATGGCCCAAGACAACAT 3'; RvIrc25, 5' CGCTTTGTTCCTTCGTTGCT 3'; FwMdj1, 5' TTTGGTGCTGCATTTGGTGG 3'; RvMdj1, 5' GGTCCAGCGCAGAGAATCTT 3'; FwTom70, 5' GCGGATGCGATGTTCGATTT 3'; RvTom70, 5' GCGTTCTTTAGCTGGT TGGG 3'; FwTom40, 5' CACTATCCCGGCCCTTTCTC 3'; RvTom40, 5' GTACCGGGATT GACCAGCTC 3'; FwTom22, 5' CCCCCAGGTAAGAGACAAACA 3'; RvTom22, 5' AGCAGTGGTGGTCAAAGTCC 3'; FwTim22, 5' TGAACGGGGTGCTGAAATGA 3'; RvTim22, 5' ACGCCATGAACAGGCCTAAA 3'; FwTim54, 5' ACTGGTGGATGCCGAG AAAG 3'; RvTim54, 5' CGAAGCGATTTGCAGGTCAG 3'; FwSam50, 5' TTGTGCCAACAC CACATTGC 3'; RvSam50, 5' TTTGTCCCTGTCTTCGCTGT 3'; FwMia40, 5' AAACTGAAG CTGGCCCTCAA 3'; RvMia40, 5' TGTTAGCGGTGGCATTGTCT 3'; FwTim23, 5' CTGCAT CCTTTGGCTGGTCT 3'; RvTim23, 5' CCGATACCAAGTCCCAGCAG 3'; FwTim50, 5' GGCTCTTAGGGTTGTGCCAT 3'; RvTim50, 5' TTTTGCCTTGGCCCTTCTCA 3'. Samples were analyzed using the CFX384 Real-time PCR detection system (Bio-Rad). The reference gene selected as internal control was *TAF10*. Data (six replicates) were analyzed as previously reported by [50].

## Membrane potential measurements

The mitochondrial membrane potential ($\Delta\psi$) across the inner mitochondrial membrane was quantified by fluorescence quenching using the voltage-dependent dye 3,3'-dipropylthiadicarbocyanine (DisC$_3$) [51]. Membrane potential buffer (0.1% [w/v] BSA, 0.6 M sorbitol, 20 mM KP$_i$, 10 mM MgCl$_2$, 0.5 mM EDTA, pH 7.2) was supplemented with DisC$_3$ and mixed with isolated mitochondria. The membrane potential was dissipated by addition of valinomycin (1 μM final concentration).

## Lipid extraction for mass spectrometry lipidomics

Mass spectrometry-based lipid analysis was performed by Lipotype GmbH (Dresden, Germany) as described [52,53]. Lipids were extracted using a two-step chloroform/methanol procedure [52]. Samples were spiked with internal lipid standard mixture containing the following phospholipid and lyso-phospholipid species: CDP-DAG 17:0/18:1, lyso-phosphatidylcholine 12:0 (LPC), lyso-phosphatidylethanolamine 17:1 (LPE), lyso-phosphatidylinositol 17:1 (LPI), lyso-phosphatidylserine 17:1 (LPS), phosphatidylcholine 17:0/14:1 (PC), phosphatidylethanolamine 17:0/14:1 (PE), phosphatidylglycerol 17:0/14:1 (PG), phosphatidylinositol 17:0/14:1 (PI), and phosphatidylserine 17:0/14:1 (PS). After extraction, the organic phase was transferred to an infusion plate and dried in a speed vacuum concentrator. 1st step dry extract was re-suspended in 7.5 mM ammonium acetate in chloroform/methanol/propanol (1:2:4, V: V:V) and 2nd step dry extract in 33% ethanol solution of methylamine in chloroform/methanol (0.003:5:1; V:V:V). All liquid handling steps were performed using Hamilton Robotics STARlet robotic platform with the Anti Droplet Control feature for organic solvents pipetting.

## MS data acquisition

Samples were analyzed by direct infusion on a QExactive mass spectrometer (Thermo Scientific) equipped with a TriVersa NanoMate ion source (Advion Biosciences). Samples were

analyzed in both positive and negative ion modes with a resolution of Rm/z = 200 = 280000 for MS and Rm/z = 200 = 17500 for MSMS experiments, in a single acquisition. MSMS was triggered by an inclusion list encompassing corresponding MS mass ranges scanned in 1 Da increments [54]. Both MS and MSMS data were combined to monitor PC as an acetate adduct and CL, PE, PG, PI and PS as deprotonated anions. MS only was used to monitor LPE, LPI, and LPS as deprotonated anions and LPC as acetate adducts [55].

### Data analysis and post-processing

Data were analyzed with in-house developed lipid identification software based on LipidX-plorer [56,57]. Data post-processing and normalization were performed using an in-house developed data management system. Only lipid identifications with a signal-to-noise ratio >5, and a signal intensity 5-fold higher than in corresponding blank samples were considered for further data analysis.

### Statistical analysis

The experiments were replicated a minimum of three times. Data is represented with means ± standard error of the mean (SEM). Statistical information of each experiment is described in the figure legends. Comparison between two groups was analyzed by a Student's t test. Significances are displayed with asterisks: not significant (n.s.) $p > 0.05$, $*p < 0.05$, $**p < 0.01$, $***p < 0.001$.

### Supporting information

**S1 Fig. Cell growth and protein import in *mas1^ts* mutant is indistinguishable from wild-type at permissive growth temperature.** (A) Growth curves of *mas1^ts* and wild-type (WT) yeast cells on respiratory growth medium (YPglycerol) at the permissive temperature 25˚C. (B) Growth curves of indicated yeast cells on respiratory growth medium (YPglycerol) at non-permissive temperature 37˚C. Dashed lines indicate time points used for import analyses (*early* mtUPR, 4 and 10 hours, *late* mtUPR 20 hours growth at elevated temperature). (C)-(H) Import kinetics of indicated radiolabeled precursor proteins into mitochondria isolated from WT or *mas1^ts* cells after growth at 23˚C. Where indicated, the membrane potential (Δψ) was dissipated prior to the import reaction. Samples were solubilized in the mild detergent digitonin and analyzed by BN-PAGE and autoradiography.
(PDF)

**S2 Fig. The presequence and carrier import pathways are increasing earliest upon mtUPR.** (A)-(D) Analysis of protein import kinetics of indicated radiolabeled precursor proteins into mitochondria isolated from wild-type (WT) or *mas1^ts* cells after induction of mtUPR for 4 hours. Hsp10 is a model substrate for the presequence import pathway and Aac2 for the carrier import pathway into the inner membrane. Where indicated, the membrane potential (Δψ) was dissipated prior to the import reaction. Samples were solubilized with the mild detergent digitonin and analyzed by BN-PAGE and autoradiography. (E) Quantification of longest import time-point displayed in (A)-(D) normalized to WT value. n = 3, data represent means ± SEM. Student's t-test was used for pairwise comparison. n.s., not significant; $*p < 0.05$; $**p < 0.01$.
(PDF)

**S3 Fig. Import of outer membrane precursors dependent on protein import machineries is increased upon mtUPR.** (A) and (B) Blue-native PAGE autoradiography of assembly of Tom40 (A) and Tom22 (B) after import into wild-type (WT) and *mas1^ts* mitochondria isolated after cell growth at non-permissive temperature for 10 hours. (C) Quantification of imports

shown in (A) and (B). The longest time point is compared between WT and *mas1*[ts]. n = 3, data represent means ± SEM. Student's t-test was used for comparison. **p < 0.01; ***p < 0.001. (PDF)

**S4 Fig. Mitochondrial protein import decreases only upon prolonged mtUPR induction (*late* mtUPR).** (A)-(C) Kinetic analysis of assembly of indicated radiolabeled precursor proteins into wild-type (WT) and *mas1*[ts] mitochondria isolated from cells after growth at 37˚C for 20 hours (*late* mtUPR). Where indicated the membrane potential ($\Delta\psi$) was dissipated prior to the import reaction. Assembled complexes were analyzed by Blue Native PAGE and autoradiography. (D) Quantification of imports shown in (A)-(C). The longest time point is compared between WT and *mas1*[ts]. n = 3, data represent means ± SEM. Student's t-test was used for comparison. **p < 0.01; ***p < 0.001. (E) Measurement of the membrane potential ($\Delta\psi$) in WT and *mas1*[ts] mitochondria 20 hours after induction of mtUPR. n = 3, data represent mean ± SEM. Student's t-test was used for comparison. n.s., not significant. (PDF)

**S5 Fig. Import into the outer membrane independent of the TOM- and SAM-translocases is not changed upon mtUPR.** (A) Kinetic analysis of Tom20 protein import, which does not depend on the TOM and SAM complexes for its assembly into the outer membrane. Samples were analyzed by BN-PAGE and autoradiography. (B) Quantification of longest time point of import and assembly of Tom20 shown in (A). n = 3, data represent mean ± SEM. Student's t-test was used for comparison. n.s., not significant. (PDF)

**S6 Fig. Gene expression and protein levels of subunits of mitochondrial protein import machineries and their assembly are not changed upon mtUPR.** (A) Gene expression analysis of representative genes of the mitochondrial import complexes by RT-qPCR after cell growth for 10 hours at 37˚C. *MDJ1*, encoding for the mitochondrial co-chaperone Mdj1 serves as a positive control for mtUPR induction; *IRC25*, control. Quantification for n = 6, data represent mean ± SEM. Student's t-test was used for comparison of WT and *mas1*[ts]. n.s., not significant; *p < 0.05; **p < 0.01; ***p < 0.001. (B) Immunoblot analysis of WT and *mas1*[ts] mitochondria isolated from cells shifted to non-permissive temperature for 10 hours. For Tim18 and Cox4 the accumulation of their precursor forms due to MPP inhibition is visible. p, precursor; m, mature protein. (C) Analysis of native protein complexes in WT and *mas1*[ts] mitochondria by Blue-native PAGE and immunodecoration. Mitochondria were isolated after growth for 10 hours at non-permissive temperature and solubilized with 1% digitonin. (PDF)

**S7 Fig. Remodelling of cardiolipin subspecies upon *early* mtUPR.** (A) Quantification of indicated cardiolipin (CL) subspecies in wild-type (WT) and *mas1*[ts] mitochondria isolated after cell growth at permissive temperature (23˚C). n = 3, data represent means ± SEM. (B)-(D) Heatmaps of distribution of different CL subspecies standardized to total CL content per sample in WT and *mas1*[ts] mitochondria isolated from cells shifted for 10 hours to non-permissive growth temperature. (C) Analysis of acyl chain length and (D) of number of double bonds. Heatmaps were generated using standardized values in mol% and thus total CL content in each individual sample was set to 100% to represent relative CL species distribution within each sample. Relative changes, scaled and centered for each CL species, are depicted. Shown are three biological replicates for each strain. (PDF)

**S8 Fig. Analyses of phospholipid biosynthesis pathways and import translocases upon mtUPR.** (A) Growth assay to test for synthetic effects of indicated deletions in wild-type (WT) and *mas1^ts^* cells. Serial dilutions were tested on respiratory (YPglycerol) and non-respiratory (YPglucose) plates and incubated at 35°C (mild mtUPR induction). (B) Analysis of translocases in WT and *mas1^ts^* mitochondria isolated from cells grown for 10 hours at non-permissive temperature. Samples were solubilized with indicated concentrations of digitonin and analyzed by Blue-native PAGE and immunodecoration. TIM22 and TIM23, translocases of the inner mitochondrial membrane. (C) Immunoblot analysis of WT and *mas1^ts^* mitochondria isolated from cells shifted to non-permissive temperature for 4 hours. (D) Analysis as in (B) using mitochondria isolated from *crd1Δ* and *crd1Δ mas1^ts^* cells grown under non-permissive temperature for 10 hours. (PDF)

**S9 Fig. Loss of cardiolipin remodelling upon mtUPR does not impact on the protein levels of import machinery subunits or the membrane potential.** (A) Protein steady state analysis of mitochondria isolated from *crd1Δ* and *crd1Δmas1^ts^* mitochondria after induction of *early* mtUPR. Samples were analyzed by SDS-PAGE and immunodecoration. p, precursor; m, mature protein. (B) Measurement of the membrane potential (Δψ) in *crd1Δ* and *crd1Δ mas1^ts^* mitochondria after induction of mtUPR. n = 3, data represent mean ± SEM. Student's t-test was used for comparison. n.s., not significant. (PDF)

**S1 Table. List of yeast strains used in this study.** (PDF)

**S2 Table. Primary antibodies used in this study.** All antisera were raised in rabbit. (PDF)

## Acknowledgments

We thank Dr. Chris Meisinger for scientific discussion of this manuscript and Dr. Doron Rapaport and Layla Drwesh for technical support and discussion.

## Author Contributions

**Conceptualization:** Daniel Poveda-Huertes, Asli Aras Taskin, Sabrina Büttner, F.-Nora Vögtle.

**Data curation:** Daniel Poveda-Huertes, Asli Aras Taskin, Ines Dhaouadi, Lisa Myketin, Adinarayana Marada, Lukas Habernig.

**Formal analysis:** Daniel Poveda-Huertes, Asli Aras Taskin, Sabrina Büttner, F.-Nora Vögtle.

**Funding acquisition:** Sabrina Büttner, F.-Nora Vögtle.

**Investigation:** F.-Nora Vögtle.

**Supervision:** Sabrina Büttner, F.-Nora Vögtle.

**Writing – original draft:** F.-Nora Vögtle.

**Writing – review & editing:** F.-Nora Vögtle.

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
