## [Editor Report · Decision Letter 0]

25 Jan 2021

Dear Dr Vögtle,

Thank you very much for submitting your Research Article entitled 'Mitochondrial protein import is stimulated upon mtUPR and requires cardiolipin remodeling' to PLOS Genetics.

The manuscript, the review history at review commons, and the extensive reply and rebuttal have been fully evaluated. Based on our evaluation, we invite you to submit a revised manuscript that incorporates the new data and key elements of the rebuttal. In the revised manuscript, we encourage you to include at least one more main figure, as we would like to see at least 4 main figures in the manuscript. We leave decision of what results should be transferred to the main figures at this time to you. We cannot, of course, promise publication at that time, as the manuscript will be reevaluated by the original reviewers. 

Should you decide to revise the manuscript for further consideration here, your revisions should address the specific points made by each reviewer. We will also require an updated "reply to reviewers" that refers to the revision, includes a detailed list of your responses to the review comments, and a description of the changes you have made in the manuscript.

If you decide to revise the manuscript for further consideration at PLOS Genetics, please aim to resubmit within the next 60 days, unless it will take extra time to address the concerns of the reviewers, in which case we would appreciate an expected resubmission date by email to plosgenetics@plos.org.

[LINK]

We are sorry that we cannot be more positive about your manuscript at this stage. Please do not hesitate to contact us if you have any concerns or questions.

Yours sincerely,

Aleksandra Trifunovic

Associate Editor

PLOS Genetics

Gregory Barsh

Editor-in-Chief

PLOS Genetics

---

## [Decision Letter · Decision Letter 1]

28 Mar 2021

Dear Dr Vögtle,

Thank you very much for submitting your Research Article entitled 'Increased mitochondrial protein import and cardiolipin remodelling upon early mtUPR' to PLOS Genetics.

The manuscript was fully evaluated at the editorial level and by independent peer reviewers. The reviewers appreciated the attention to an important problem, but raised some substantial concerns about the current manuscript. Based on the reviews, we will not be able to accept this version of the manuscript, but we would be willing to review a much-revised version. We cannot, of course, promise publication at that time.

If you decide to revise the manuscript for further consideration at PLOS Genetics, please aim to resubmit within the next 60 days, unless it will take extra time to address the concerns of the reviewers, in which case we would appreciate an expected resubmission date by email to plosgenetics@plos.org.

[LINK]

We are sorry that we cannot be more positive about your manuscript at this stage. Please do not hesitate to contact us if you have any concerns or questions.

Yours sincerely,

Aleksandra Trifunovic

Associate Editor

PLOS Genetics

Gregory Barsh

Editor-in-Chief

PLOS Genetics

Reviewer's Responses to Questions

**Comments to the Authors:**

Reviewer #1: This is a revised version of the manuscript previously submitted to PROS Genetics. The authors added substantial amounts of new results, which could strengthen and improve the manuscript. Their observation on the effects of defective MPP processing in mas-ts mutant mitochondria is very interesting and would offer an important basis for future studies on cellular responses to the defects in the mitochondrial protein import and related processes. Therefore, I have no objection against the importance of the present work at the level of descriptive reports. My still-remaining concern is the authors’ interpretation that the defective MPP processing causes the increase in the CL level, and that the increased CL level enhances the mitochondrial protein import efficiencies. I do not see that the logic behind this interpretation was experimentally and convincingly demonstrated here. Therefore, I suggest the authors distinguish what was experimentally demonstrated clearly and what is still at the level of speculation throughout the manuscript. Here are the points I suggest the authors consider.

(1) It is understandable that the deletion of the CRD1 gene or decrease in the CL level caused import defects because the decreased CL level should affect structures and functions of inner membrane protein complexes in general. However, this does not mean that the reverse relationship is expected, that is, it was not demonstrated that the increased CL level directly enhanced the protein import efficiencies. Negative synergetic effects of the deletion of TAZ1 and mas1-ts mutation may support the idea that accumulated presequence-containing precursor proteins could cause structural perturbation of the inner membrane, but does not again support the reverse effects that the increased CL level would enhance the protein import efficiencies. In this sense, the authors’ new findings that the increase in the levels of Ups1 and Pgs1 could explain the increased CL level are important. I suggest that the authors should increase the levels of Ups1 and Pgs1 in wild-type cells, which would increase the CL level, and then test the possible enhancement of the protein import.

(2) The authors found that the steady-state level of mtHsp10 increased upon defective MPP processing, which may complement the results of the in vitro import. However, the level of mtHsp10 has a problem that mtHsp10 is a heat-shock protein. Therefore, the level of mtHsp10 should be tested after changing its promoter with an unrelated one to minimize the effects of the heat-shock response but to reflect import efficiency in vivo. An arising related question is what about the protein levels of mitochondrial proteins for the other import pathways are. It looks that Por1 did not change its levels between WT and mas1-ts cells. How about Tim9 and AAC?

(3) It is still not clear how the increased CL level further increases the import ability of mitochondrial import machineries, although this may be beyond the scope of this study. The new results on the digitonin extractability of the TIM22 complex are interesting, but should not be used for over-interpretation. For example, what about the TOM complex and SAM complex?

Reviewer #2: The authors improved their text, clarified unclear points, and added some more experiments. However, as indicated in detail below, the relevance of the apparent changes in the cardiolipin (CL) levels and composition to difference in protein import efficiency is still not convincing. Some of the experiments lack crucial controls and/or does not support the causative effect of altered CL levels on protein import.

Main comments:

1. Fig. 2B: The levels of CL in the mas1-ts strain after 4 hrs at 37°C are similar to those of WT without shift to 37°C. Thus, if the shift to 37°C under these conditions should represent “stress induction”, I cannot detect any compensation changes in the CL levels as compared to normal non-stressed conditions that should support enhanced import efficiency.

2. Fig. 2C and lines 214-219: The increase in the levels of CDP-DAG is observed also for WT cells after 2 hrs at 37°C. In addition, the change in the CDP-DAG for the mas1ts strain within the first two hrs at 37°C is only 0.35 pmol whereas the increase in the levels of CL between 2 to 4 hrs is about 30 pmol (100 fold). The authors describe these changes as “This suggests that cells increase the biosynthesis of CL upon mtUPR and rapidly convert its precursor into CL”. However, if this is the case, why there is no change in CL amount also after 2 hrs at 37°C?

In addition, according to the authors claims, the CL levels should be even higher after 10 hrs and then decline again towards the 20 hrs stage (“late stages”). However, such data is not provided as Fig. 2B shows data only for 2 and 4 hrs.

3. Fig. 2E and S7B-D: It is not clear what this panel represents. Are those absolute numbers or differences to cells that were not shifted to 37°C? What is the scale of -2 to +2? In addition, it seems that the trend in the relatively short species of CL is different between 4 to 10 hrs (compare Figs. 2E to S7B) although both conditions should support more import.

4. Fig. 3E: The figure shows apparent higher levels of the TIM22 complex in the control organelles upon solubilization with 0.5% Digitonin. However, without appropriate control (like SDS-PAGE analysis followed by Western blotting) of the very same samples side-by-side we do not know if this is a difference in solubilization or the outcome of loading variable amounts. Generally, I am not sure that detecting less material under these conditions can indicate higher stability of the monitored complex or represents an overall different behavior of the solubilized membrane.

5. The authors write in the M&M section that import of Porin and Tom20 was analyzed after solubilization with 0.3% [w/v] digitonin. On the other hand, the authors claimed that lower conc. of digitonin (like 0.5%) can extract less complexes from the tas1ts strain (Fig. 3E). Hence, the authors should comment whether differences that are observed under these conditions indeed reflect the variation in import efficiency or in solubilization efficiency.

Minor issues:

a. Fig. 1C: what are the two bands at about 100 kDA? These complexes, which might represent the Tim9/10 and/or Tim9/10/12 complexes, are formed faster in WT rather than slower. The authors should comment on this in the text.

b. Line 171: the authors write “our conditional mas1ts mutant allows mild and reversible stress induction”. However, they did not show that cells that were initially transferred to the non-permissive temperature and then back to permissive temperature have wild type-like import behavior. Hence, I suggest that they will rephrase their claim.

c. Fig. 2G: I do not see a difference in the levels of Pgs1 (40 µg) between WT and mas1ts strains. The authors should change their text or show more convincing data.

d. Figs. 3C-D and S1A&B: Do these curves represent continuous monitoring of the OD? If yes, this should be indicated in the M&M section or in the legends. If not, the values of the distinct time points should be indicated.

e. Line 784: should be “in (B)-(E)”.

f. Legend to Fig. S7: The description of panels C and D are referred to panels B and C.

g. Fig. S8B: I do not see a difference in the behavior of the TIM23 complex between the two strains. The authors should change their text or show more convincing data.

Reviewer #3: The authors have revised their work in light of the previous comments. I am very happy to see that they addressed all the issues raised and provided a wealth of additional experiments which make the manuscript much stronger and clarified further several points in their work. One of the points I raised (major point 3 on levels of CDP-DAG) is still in my opinion not very convincingly addressed. However, as the authors really have substantiated all other supporting evidence with additional work I am happy to recommend acceptance and publication of this very interesting work.

**Have all data underlying the figures and results presented in the manuscript been provided?**

Reviewer #1: Yes

Reviewer #2: Yes

Reviewer #3: Yes

PLOS authors have the option to publish the peer review history of their article (what does this mean?). If published, this will include your full peer review and any attached files.

Reviewer #1: No

Reviewer #2: No

Reviewer #3: No

---

## [Decision Letter · Decision Letter 2]

29 May 2021

Dear Dr Vögtle,

Thank you very much for submitting your Research Article entitled 'Increased mitochondrial protein import and cardiolipin remodelling upon early mtUPR' to PLOS Genetics.

The manuscript was fully evaluated at the editorial level and by independent peer reviewers. The reviewers appreciated the effort put into revised manuscript as well as important topic that it covers.  Nevertheless, reviewer 1 still raised some concerns that we ask you to address in a revised manuscript. We therefore ask you to modify the manuscript according to the review recommendations. 

[LINK]

Yours sincerely,

Aleksandra Trifunovic

Associate Editor

PLOS Genetics

Gregory Barsh

Editor-in-Chief

PLOS Genetics

Reviewer's Responses to Questions

**Comments to the Authors:**

Reviewer #1: This is the second revised version of the manuscript submitted to PROS Genetics. I am a bit disappointed to see that the authors have not made any further efforts to clear my remaining concerns this time.

(1) My major concern in the last round of revision was …“my still-remaining concern is the authors’ interpretation that the defective MPP processing causes the increase in the CL level, and that the increased CL level enhances the mitochondrial protein import efficiencies. I do not see that the logic behind this interpretation was experimentally and convincingly demonstrated here.” The authors’ answer to this concern was.. “we suggest that this concomitant CL remodelling might play a role in the observed enhanced protein import. However, we do not claim that the increased import is exclusively caused by CL remodelling.” However, I do not see the authors’ claim of “this concomitant CL remodelling might play a role in the observed enhanced protein import”, even if this is just a suggestion, is based on the experimental results, and thus inappropriate here.

For example, the authors stated “if modulation of cardiolipin upon mtUPR and the concomitant changes in the lipid environment around the translocases are a requirement for the observed stimulation of protein import upon mtUPR” (page 12), but the performed experiment was designed to see the effects of crd1∆ mutation in wild type and mas1-ts strains, confirming that the CL level decrease would abrogate the import efficiencies, but not demonstrating that the CL increase promotes the import efficiencies. The mechanism of even the increase in the levels of Ups1 and Pgs1 is still not clear. Therefore, the authors’ conclusion of “this modulation in the lipid environment likely stabilizes the mitochondrial import complexes thereby increasing the protein import capacity into the organelle under stress conditions” (page 12) was not experimentally supported. Therefore, the authors had better withdraw this “suggestion” from the manuscript if the manuscript stays as it stands.

(2) I still do not understand that why mas1-ts mutant is more suitable for analyzing the physiologically relevant stress for mitochondria. Accumulation of non-cleaved precursor proteins in the matrix could be harsher to mitochondria than mitochondria with slightly reduced membrane potential, and the inhibition of presequence-cleavage would not occur under physiological conditions. In addition, even with a control of wild-type cells at elevated temperature, the effects of the shift to non-permissive temperature could give profoundly complicated synthetic effects with mas1-ts mutation to Hsp10 as well as to the cell. Therefore, the authors had better soften the description on the disadvantage of the previous studies using the conditions of the reduced membrane potential. The authors can simply expect different effects arising from the different stress conditions for mitochondria.

In summary, simultaneous observation of the effects on the import efficiencies and lipid compositions under the conditions of the mas1-ts mutation are interesting, but mechanistic reasoning of this observation has not been experimentally or properly tested. Therefore, this manuscript remains at a descriptive level, and interpretation of the observation that the defective MPP processing causes the increase in the CL level, and that the increased CL level enhances the mitochondrial protein import efficiencies is not logically appropriate. The authors had better emphasize the significance of the observation of the two phenomena, the change in lipid composition and enhancement of protein import efficiencies, that were previously thought to be unrelated. I think this is the point that is novel enough to be published.

Reviewer #2: The authors addressed my concerns and the manuscript can be published.

**Have all data underlying the figures and results presented in the manuscript been provided?**

Reviewer #1: Yes

Reviewer #2: Yes

PLOS authors have the option to publish the peer review history of their article (what does this mean?). If published, this will include your full peer review and any attached files.

Reviewer #1: No

Reviewer #2: No

---

## [Editor Report · Decision Letter 3]

11 Jun 2021

Dear Dr Vögtle,

We are pleased to inform you that your manuscript entitled "Increased mitochondrial protein import and cardiolipin remodelling upon early mtUPR" has been editorially accepted for publication in PLOS Genetics. Congratulations!

Yours sincerely,

Aleksandra Trifunovic

Associate Editor

PLOS Genetics

Gregory Barsh

Editor-in-Chief

PLOS Genetics

Comments from the reviewers (if applicable):

**Data Deposition**

http://datadryad.org/submit?journalID=pgenetics&manu=PGENETICS-D-21-00040R3

**Press Queries**

---

## [Editor Report · Acceptance letter]

29 Jun 2021

PGENETICS-D-21-00040R3 

Increased mitochondrial protein import and cardiolipin remodelling upon early mtUPR 

Dear Dr Vögtle, 

We are pleased to inform you that your manuscript entitled "Increased mitochondrial protein import and cardiolipin remodelling upon early mtUPR" has been formally accepted for publication in PLOS Genetics! Your manuscript is now with our production department and you will be notified of the publication date in due course.

With kind regards,

Zsofi Zombor

PLOS Genetics

On behalf of:
